# An Integrated Cognitive Remediation and Recovery-Oriented Program for Individuals with Bipolar Disorder Using a Virtual Reality-Based Intervention: 6- and 12-Month Cognitive Outcomes from a Randomized Feasibility Trial

**DOI:** 10.3390/bs15040452

**Published:** 2025-04-01

**Authors:** Alessandra Perra, Mauro Giovanni Carta, Diego Primavera, Giulia Cossu, Aurora Locci, Rosanna Zaccheddu, Federica Piludu, Alessia Galetti, Antonio Preti, Valerio De Lorenzo, Lorenzo Di Natale, Sergio Machado, Antonio Egidio Nardi, Federica Sancassiani

**Affiliations:** 1Department of Medical Sciences and Public Health, University of Cagliari, 09124 Cagliari, Italy; alessandra.perra@unica.it (A.P.); giulia.cossu@unica.it (G.C.); auroralocci@gmail.com (A.L.); rosannazaccheddu@gmail.com (R.Z.); federica.piludu@outlook.it (F.P.); alessia.galetti@unica.it (A.G.); federicasancassiani@yahoo.it (F.S.); 2Department of Neuroscience, University of Turin, 10125 Turin, Italy; antonio.preti@unito.it; 3Department of Systems Medicine, University of Tor Vergata, 00133 Rome, Italy; valerio.delorenzo@gmail.com; 4Cerebrum VR Society, 00185 Rome, Italy; lorenzodinatale86@gmail.com; 5IDEGO Digital Psychology Society, 00133 Rome, Italy; 6Centre of Neuroscience, Neurodiversity Institute, Queimados 26325-010, Brazil; secm80@gmail.com; 7Institute of Psychiatry, Federal University of Rio de Janeiro, Rio de Janeiro 22290-140, Brazil; antonioenardi@gmail.com

**Keywords:** bipolar disorder, cognitive remediation, virtual reality, psychiatric rehabilitation

## Abstract

Introduction: Achieving long-term impacts from cognitive remediation (CR) interventions is a key goal in rehabilitative care. Integrating virtual reality (VR) with psychoeducational approaches within CR programs has shown promise in enhancing user engagement and addressing the complex needs of individuals with bipolar disorder (BD). A previous randomized controlled crossover feasibility trial demonstrated the viability of a fully immersive VR-CR intervention for BD, reporting low dropout rates, high acceptability, and significant cognitive improvements. This secondary analysis aimed to evaluate the stability of these outcomes over time. Methods: This paper presents a 6- to 12-month follow-up of the initial trial. Secondary cognitive outcomes were assessed, including visuospatial abilities, memory, attention, verbal fluency, and executive function, using validated assessment tools. Statistical analyses were conducted using Friedman’s test. Results: A total of 36 participants completed the 6- to 12-month follow-up. Overall, cognitive functions showed a trend toward stability or improvement over time, except for visuospatial and executive functions, which demonstrated inconsistent trajectories. Significant improvements were observed in language (*p* = 0.02). Conclusion: This study highlights the overall stability of cognitive functions 12 months after a fully immersive VR-CR program for individuals with BD. To sustain long-term clinical benefits, an integrated approach, such as incorporating psychoeducational strategies within cognitive remediation interventions, may be essential. Further follow-up studies with control groups and larger sample sizes are needed to validate these findings.

## 1. Introduction

Bipolar disorder (BD) is one of the leading causes of global disability ([15]; [58]; [55]). It is frequently associated with cognitive impairment (CI), affecting a significant proportion of individuals diagnosed with BD ([20]). CI is strongly linked to poor recovery in personal, social, and occupational functioning ([6]; [23]). Therefore, improving CI is a critical therapeutic target in BD treatment ([36]; [52]). Given its multifactorial nature, CI in BD requires integrated and complex interventions for effective management ([44]).

Cognitive remediation (CR) is a well-established intervention aimed at enhancing cognitive functioning and improving daily life in individuals with mental health conditions, including BD. Importantly, CR promotes long-term functional recovery by addressing cognitive deficits ([53]). Traditionally, validated CR interventions include paper-based and computerized methods ([16]). Recent studies have demonstrated that conventional CR approaches for individuals with BD lead to significant cognitive improvements ([34]; [4]; [50]). Functional remediation (FR) is an adaptation of standard CR that was specifically designed for BD. This approach addresses cognitive deficits within an ecological framework, integrating psychoeducation on neuropsychological impairments and their impact on daily life ([51]; [45]; [41]). FR facilitates the incorporation of various rehabilitative strategies within CR sessions, enabling the development of comprehensive interventions aligned with the biopsychosocial model ([42]; [26]). This integrative approach is consistent with the latest framework for designing complex interventions ([48]).

Although further high-quality studies on BD are needed, the demonstrated efficacy of CR highlights is potential as a valuable therapeutic tool for improving CI, which, in turn, impacts psychosocial functioning ([53]; [45]).

However, studies on the effectiveness of CR in BD face several limitations. These include high variability in cognitive and psychosocial outcomes, methodological weaknesses such as small sample sizes, study designs that lack adequate statistical power, higher dropout rates with paper-and-pencil methods compared to other populations, and the absence of a structured framework, which limits intervention reproducibility ([53]; [42]). Additionally, the lack of long-term follow-up studies prevents a comprehensive evaluation of CR’s sustained effects, despite its critical role in rehabilitation ([61]; [19]). These factors hinder a clear understanding of the variables influencing treatment outcomes. Naturally, the results may be affected by the study design, methodological choices, and intervention quality. Furthermore, characteristics related to the intervention type and implementation also play a crucial role ([60]; [53]). In particular, CR interventions require a robust theoretical framework to ensure a structured and targeted approach that aligns with specific clinical needs ([42]; [48]).

A well-defined session structure enhances intervention reproducibility. Research involving other populations, such as individuals with schizophrenia, has highlighted the pivotal role of the therapist in CR sessions. Strategies that facilitate the transfer of cognitive improvements to real-world contexts are essential for aligning with personal recovery goals. These strategies include discussion-based sessions, role playing, and out-of-session tasks such as homework assignments ([53]). Certain CR methodologies emphasize the integration of these strategies to foster metacognitive development ([17]).

Moreover, studies suggest that therapist involvement is critical in promoting adherence and reducing dropout rates in CR interventions ([32]). Additionally, intensive cognitive training (massed practice) has been linked to greater neural network engagement and improved cognitive outcomes ([56]). A particularly relevant finding in BD research is the presence of a dose–response relationship, where a higher number of CR sessions correlates with greater cognitive and functional improvements. Completing at least 20 sessions has been associated with substantial treatment benefits. In schizophrenia research, an optimal CR session frequency has been estimated at 2.5 sessions per week ([57]), although maintaining at least two sessions per week remained effective ([43]).

Beyond methodological considerations, recent years have seen a growing interest of immersive technologies, such as virtual reality (VR), in rehabilitative interventions ([54]; [27]). The integration of fully immersive VR-based CR may enhance user engagement, particularly in individuals with BD, and facilitate the transfer of learned skills to daily life ([28]; [41]). Given BD’s characteristic exploratory behavior and hyperactivity ([11]), VR-based interventions could provide an engaging and playful yet effective rehabilitation approach.

To explore these aspects, a randomized controlled crossover feasibility trial was conducted following established CR methodologies ([40]). The primary objective was to evaluate feasibility outcomes, including acceptability, tolerability (dropout rates), satisfaction, and potential side effects. The secondary objective was to gather preliminary data on effectiveness. Specifically, the study assessed cognitive domains (attention, memory, executive functions, language, and visuospatial skills) and psychosocial variables relevant to BD, such as depressive and anxiety symptoms, quality of life, biological rhythms, and emotional awareness. Cognitive impairment in BD significantly impacts quality of life and contributes to biological and social rhythm dysregulation ([14], [12], [13]).

The post-treatment results ([41]) demonstrated the feasibility of a fully immersive VR-CR intervention for BD, with low dropout rates, high acceptability, and high satisfaction levels, consistent with previous studies (Tsapekos et al., 2022 2023. Additionally, the preliminary evidence suggested significant cognitive and psychosocial improvements following the VR-CR intervention.

### Aim

This study represents a secondary analysis of a previously conducted randomized controlled crossover feasibility trial. The objective was to evaluate the stability of the cognitive function outcomes at 6- and 12-month follow-ups after completing a fully immersive VR-CR program that incorporated an integrated psychoeducational approach.

## 2. Materials and Methods

### 2.1. Study Design

This study was a 6- and 12-month follow-up of a previous randomized controlled crossover feasibility trial ([40]). Due to the crossover design, a control group was not available for the 6- and 12-month follow-ups. The original study adhered to the reporting standards outlined in the CONSORT extension for feasibility studies ([24]) and was registered on ClinicalTrials.gov in September 2021 under protocol number NCT05070065.

### 2.2. Participants and Simple Size Considerations

Among the 50 individuals with BD who participated in the previous study ([41]), 36 were evaluated 6 and 12 months after the intervention (t2, t3), as shown in the flow diagram (Figure 1). The participants were recruited from the Consultation Psychiatry and Psychosomatic Center at the University Hospital of Cagliari. The dropout rate was 22% post-treatment, as previously reported ([41]). All inclusion and exclusion criteria details are provided elsewhere ([41]). Briefly, the inclusion criteria were individuals aged 18 to 75 years, a diagnosis of BD, and inclusion of both sexes. The exclusion criteria included individuals experiencing manic/depressive episodes, those diagnosed with epilepsy, or those with severe ocular conditions due to the potential risks associated with VR stimulation. Regarding the sample size, an initial estimate targeted 60 participants to assess the feasibility outcomes and provide preliminary efficacy data for the VR intervention. However, given the limited evidence in this research area, a precise sample size calculation was not feasible ([39]; [31]). The final sample comprised 50 participants due to the early cessation of the enrollment caused by the COVID-19 pandemic. The primary aim remained assessing the feasibility and gathering preliminary efficacy data, with a more detailed sample size calculation planned for the efficacy study based on Phase II data ([1]).

### 2.3. Outcomes and Data Collection

The previous study, a Phase 2 feasibility study, primarily assessed feasibility, while the secondary outcomes included a preliminary evaluation of efficacy in terms of cognitive and psychosocial variables (personal and social functioning, levels of anxiety, depressive symptoms, alexithymia, quality of life, and biological and social rhythms). For the present study, the cognitive outcomes, previously considered secondary, were the primary focus. The cognitive functions assessed included visuospatial abilities, attention, memory, verbal and semantic fluency, and executive function. Assessments were conducted at baseline (pre-treatment), immediately post-treatment, and after 6 and 12 months using validated and standardized cognitive assessment tools in their Italian versions. These tools are widely used in psychiatric literature for evaluating cognitive deficits in BD ([22]; [35]), allowing for a comprehensive analysis of individual performance across cognitive subdomains to support personalized cognitive training.

-Visuospatial function: Rey Figure Test ([7]);-Attention and immediate recall: Matrix Test ([49]) and Rey’s Words Test ([9]);-Attention function: Forward Digit Span ([38]; [5]) and Trail Making Test, Part A ([49]; [30]);-Memory function: Rey’s Words Test Delayed Recall ([9]), Test of the Tale ([10]), and Backward Digit Span ([38]; [5]);-Language function: Phonological and Semantic Verbal Fluency Test, both versions ([10]; [37]);-Executive function: Digital Symbol Substitution Test ([3], [2]), Trail Making Test, Part B ([2]), Stroop Test ([8]), Frontal Assessment Battery (FAB) ([21]), and Cognitive Estimates Test (CET) ([47]; [18]).

### 2.4. Intervention

The experimental group underwent the fully immersive VR-CR intervention using “CEREBRUM” software (version 3.0.1), developed by the Cerebrum VR Society (Rome, Italy). This software is among the latest tools in psychiatric rehabilitation that was specifically designed for CR. Created by mental health professionals, CEREBRUM is compatible with the “Oculus Quest” VR headset, which bears the CE mark, ensuring compliance with European safety, health, and environmental protection standards. The software enables training in various cognitive functions through exercises of increasing difficulty. Within the immersive virtual environment, the participants engaged with scenarios simulating both domestic and urban life. The intervention lasted three months, with 2 sessions per week, totaling 24 sessions.

To ensure clarity and facilitate replication, the logical framework guiding the study’s objectives was outlined in previous publications ([40], [41]). The intervention was conceptualized based on the specific population’s needs, targeting cognitive difficulties relevant to BD. Clear intervention objectives were defined, and the chosen session methodology was structured to support these outcomes. The intervention followed a bio-psycho-social theoretical model, integrating various techniques to address multiple variables, from cognitive to psycho-social aspects, ultimately promoting the generalization of trained skills and long-term effects. Previous publications provide a detailed explanation of the session methodologies designed to enhance the reproducibility of the intervention.

### 2.5. Statistical Analysis

The data were analyzed using SPSS software (version 21). Descriptive statistics (means ± SD) were calculated for each outcome at T0, T1, T2, and T3. Cognitive variables scores were adjusted using normative data corrected for age and education level, with equivalent scores (0–4) used when available. The sample distribution was assessed using the Kolmogorov–Smirnov test for normality. As the data did not follow a normal distribution across the different time points, the Friedman test was applied.

## 3. Results

Thirty-four participants in the experimental VR-CR group completed both the 6- and 12-month follow-up assessments post-intervention and were included in the analysis. The dropout rate was 32% at both follow-ups. Table 1 presents descriptive statistics for the outcomes assessed at T0, T1, T2, and T3. According to the Kolmogorov–Smirnov test for distribution analysis, only the Rey Immediate Figure, Matrix Test, Substitute Symbol Digit Test, and Test of Tale followed a normal distribution across all time points. All the other tests did not follow a normal distribution at the four evaluation time points.

Overall, the cognitive variables showed a trend toward stability or improvement over time (*p* > 0.05). The Verbal Phonological/Semantic Test (language function) demonstrated significant improvement (*p* < 0.05) over time (Table 2). However, the Cognitive Estimation Test (executive functions) and the Rey Immediate Figure (visuospatial function) exhibited less clear patterns, with significant changes over time (*p* = 0.00; *p* = 0.01) but no consistent directional trend (Table 1 and Table 2).

## 4. Discussion

This study aimed to provide preliminary evidence of the sustained improvement in cognitive functions among individuals with BD who participated in a fully immersive VR-CR program when assessed at 6 and 12 months post-intervention. Consistent with our hypothesis, the findings indicated overall stability in cognitive function improvements over time. Additionally, the long-term dropout rate was in line with that of previous studies ([61]). For this study, we selected software that allowed for the training of various cognitive functions in everyday life contexts. This decision was based on an analysis of the specific needs of individuals with BD ([11]; [41]) and previous literature ([17]; [52]). Reeder et al., 2017; Freeman et al., 2019). A structured CR intervention methodology was also adopted, aligning with biopsychosocial models and recovery-oriented approaches ([40]; [46]).

The intervention consisted of two sessions per week over a three-month period. Strategies were implemented to facilitate the transfer of skills to daily life, including personalized homework assignments. Additional techniques, such as massed practice, metacognitive strategies, mindfulness exercises, and psychoeducation, were incorporated to reinforce the skills being developed. The results of this study, consistent with both the intervention methodology and theoretical framework, demonstrated good feasibility over time and provided preliminary evidence supporting the stability of cognitive function improvements, particularly in semantic and verbal language functions. Clinically, these findings suggest that VR-based CR interventions could be integrated into existing treatment plans for individuals with BD to enhance cognitive functioning and support long-term recovery. Clinicians might consider using VR-CR as a supplementary tool to traditional cognitive therapies, particularly for improving cognitive skills in real-life contexts. Given the observed stability of improvements at 6 and 12 months, VR-CR could help individuals with BD maintain cognitive gains and apply them to everyday activities, ultimately improving personal and social functioning. Moreover, the inclusion of psychoeducation and metacognitive strategies could enhance the therapeutic process by addressing both the cognitive and emotional aspects of functioning.

However, there were several limitations to consider in this study. First, due to the crossover design, it was not possible to compare the experimental group with a control group. While this design increased the statistical power and ensured that all the participants received the experimental treatment, it limited the ability to evaluate the long-term effects in a control group. Additionally, as this was a feasibility study, the results—despite the use of multivariate analysis to compare groups over time—require confirmation in future research. Despite these limitations, feasibility studies are valuable in assessing whether an intervention shows preliminary evidence of improving clinical and psychosocial outcomes. Such findings help refine intervention strategies and inform future research on the development of complex psychiatric rehabilitation programs using VR. This study provides valuable insights that should be further explored in effectiveness trials with larger samples and more advanced statistical analyses. An important direction for future research is to examine the impact of neurological diseases, such as multiple sclerosis, and the role of co-occurring anxiety symptoms on cognitive functioning in BD ([33]; [29]; [25]). In the present study, this analysis was not possible due to the low prevalence of these neurological comorbidities in the sample, which did not allow for the identification of specific subgroups. Additionally, future follow-up studies, including a control group, are necessary to confirm the long-term effects on cognitive and psychosocial functioning.

## 5. Conclusions

The fully immersive VR-CR program appears to be a promising tool for individuals with BD, with potential long-term benefits for visuospatial abilities, verbal and semantic fluency, and executive functioning. To ensure the sustained impact of these clinical outcomes, an integrated approach is crucial, incorporating psychoeducational strategies within cognitive remediation interventions.

A key strength of this study is the use of advanced VR technology, which enhances engagement in psychosocial rehabilitation programs. Furthermore, the study highlights long-term outcomes, a critical aspect for the sustainability of mental health rehabilitation interventions. These findings, along with previously published data, underscore the importance of further investigation given the program’s feasibility and the preliminary evidence of post-treatment stability at 6 and 12 months.

## Figures and Tables

**Figure 1 behavsci-15-00452-f001:**
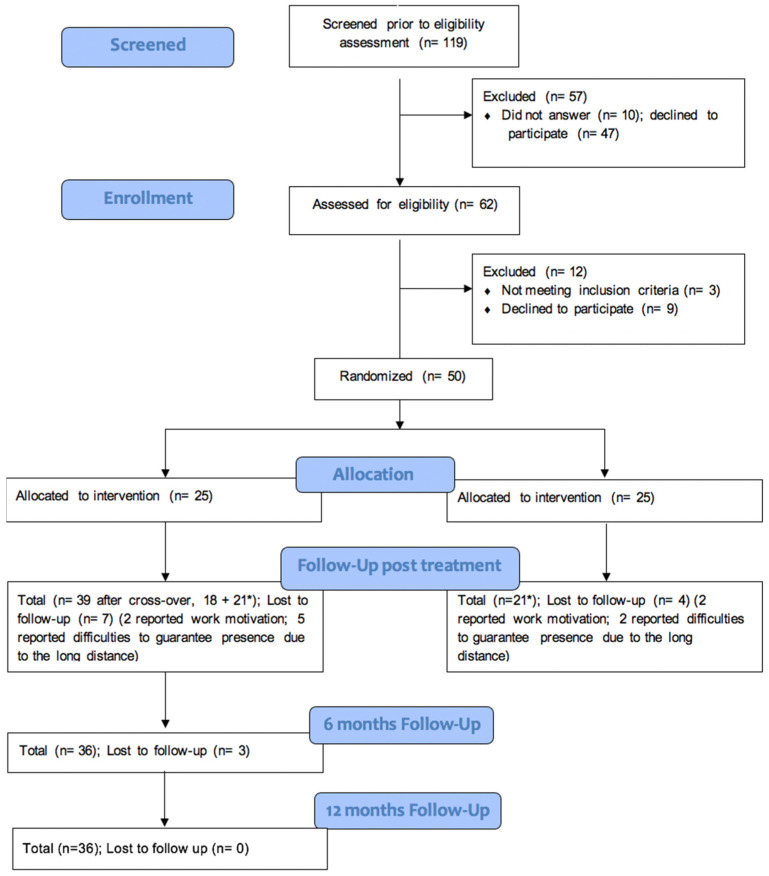
CONSORT extension flow diagram for feasibility study. * refers to 21 participants that, due to the crossover methodology, were initially part of the control group. After the washout period they became part of the intervention group.

**Table 1 behavsci-15-00452-t001:** Cognitive test—descriptive analysis (mean and standard deviation).

VR/CR GROUP (N = 36)				
Test	T0	T1	T2	T3
Fig. Rey Immediate (Vis. Sp.)	29.04 ± 8.60	30.68 ± 6.88	28.11 ± 9.13	28.83 ± 8.73
Matrix (Attent.)	1.94 ± 1.37	2.36 ± 1.47	1.94 ± 1.41	1.88 ± 1.40
Digit Span Forward (Attent.)	2.86 ± 1.47	2.75 ± 1.50	2.66 ± 1.53	2.75 ± 1.46
Rey’s Words Immediate (Attent.)	2.38 ± 1.62	3.02 ± 1.50	2.66 ± 1.80	2.72 ± 1.75
TMT-A (Attent.)	2.88 ± 1.21	3.02 ± 1.42	3.11 ± 1.18	3.16 ± 1.20
Rey’s Words Delayed (Memory)	2.25 ± 1.57	2.89 ± 1.55	2.70 ± 1.72	2.75 ± 1.67
Digit Span Backward (Memory)	1.97 ± 1.64	2.16 ± 1.68	2.15 ± 1.52	2.61 ± 1.49
Test of Tale (Memory)	2.16 ± 1.44	2.77 ± 1.17	2.25 ± 1.15	2.52 ± 1.18
Verbal Phonological Test (Leng.)	2.63 ± 1.43	3.05 ± 1.28	3.13 ± 1.37	3.11 ± 1.42
Verbal Semantic Test (Leng.)	2.63 ± 1.31	3.22 ± 1.14	2.97 ± 1.34	2.97 ± 1.34
Substit. Digit Symbol (Ex. Fun.)	35.9 ± 13.97	38.55 ± 11.89	37.81 ± 14.45	38.56 ± 14.71
TMT-B (Ex. Fun.)	2.91 ± 1.29	3.02 ± 1.20	3.08 ± 1.36	3.02 ± 1.34
Stroop Test Time (Ex. Fun.)	2.61 ± 1.60	2.94 ± 1.45	2.83 ± 1.59	2.75 ± 1.61
FAB (Ex. Fun.)	15.0 ± 3.20	15.58 ± 2.66	15.27 ± 2.95	15.30 ± 2.92
Cognitive Estimation Test (Ex. Fun.)	1.94 ± 1.47	2.80 ± 1.09	1.88 ± 1.36	1.88 ± 1.34

**Table 2 behavsci-15-00452-t002:** Cognitive test—repeated measures Friedman’s test analysis.

Test	χ²	*p*
Fig. Rey Immediate (Vis. Sp.)	10.61	0.01
Matrix (Attent.)	5.15	0.16
Digit Span Forward (Attent.)	0.12	0.98
Rey’s Words Immediate (Attent.)	6.12	0.10
TMT-A (Attent.)	2.77	0.42
Ray’s Words Delayed (Memory)	7.10	0.06
Digit Span Backward (Memory)	4.50	0.21
Test of Tale (Memory)	3.68	0.29
Verbal Phonological Test (Language)	9.10	0.02
Verbal Semantic Test (Language)	9.22	0.02
Substit. Digit Symbol (Ex. Fun.)	0.66	0.88
TMT-B (Ex. Fun.)	2.20	0.53
Stroop Test Time (Ex. Fun.)	1.35	0.71
FAB (Ex. Fun.)	1.09	0.77
Cognitive Estimation Test (Ex. Fun.)	16.47	0.00

## Data Availability

Data sharing is not applicable to this article.

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
