# Peer review of "An Integrated Cognitive Remediation and Recovery-Oriented Program for Individuals with Bipolar Disorder Using a Virtual Reality-Based Intervention: 6- and 12-Month Cognitive Outcomes from a Randomized Feasibility Trial"

_behavsci, 2025, doi:10.3390/bs15040452_

Round 1
Reviewer 1 Report
Comments and Suggestions for Authors
Well-conducted study with current methodology clearly guided by past studies.
Some of the technical terms such as the significance of "CE-marked" headset and the "special emphasis" placed on outlining the logical framework could be elaborated to help readers apprehend the importance of indicating these.
The 2nd paragraph of the results section indicating "...overall trend toward stability or improvement ...except for the Cognitive 203 Estimation Test (executive functions) ..., despite showing significant..." may require further rephrasing. The use of "except" and "despite" may be re-considered in these sentences. In addition, while improvement in most of the outcomes were shown between T0 and T1, the sustainability of the improvement may need to be re-phrased.
Otherwise, the rigour of this study was similar to prior studies and has been an easy to read and structured paper.
Author Response
Review 1
Well-conducted study with current methodology clearly guided by past studies.
Thank you for your feedback and the time spent in helping improve the article. All modifications are highlighted in red in the text.
Some of the technical terms such as the significance of "CE-marked" headset and the "special emphasis" placed on outlining the logical framework could be elaborated to help readers apprehend the importance of indicating these.
We appreciate the comment, as it allows us to clarify certain aspects that are sometimes taken for granted but are important to make explicit for effective information dissemination.
The 2nd paragraph of the results section indicating "...overall trend toward stability or improvement ...except for the Cognitive 203 Estimation Test (executive functions) ..., despite showing significant..." may require further rephrasing. The use of "except" and "despite" may be re-considered in these sentences. In addition, while improvement in most of the outcomes were shown between T0 and T1, the sustainability of the improvement may need to be re-phrased.
Thank you for the suggestion. We have revised the wording in the results section, highlighted in red. We agree on the need to rephrase the reported findings. It is important to note that the primary outcome of the original study was to assess feasibility, while evaluating preliminary evidence was only a secondary outcome. These data suggest the importance of further investigation in this direction, given the feasibility outcomes and the preliminary post-treatment evidence at 6 and 12 months (this phrase was also added in the conclusion section).
Otherwise, the rigour of this study was similar to prior studies and has been an easy to read and structured paper.
Thank you for these considerations.
Reviewer 2 Report
Comments and Suggestions for Authors
This is an interesting study examining the long-term impact of Cognitive Remediation (CR) interventions in BD patients. Some points for a major revision are below:
How was the sample size estimated? Please add in the Methods section.
Authors should discuss what that they did not include anxiety and depression questionnaires which heavily influence the neuropsychological performance of the participants with BD (for a relevant methodological comment on this to use about high anxiety levels: https://europepmc.org/article/med/28803143 as well as for existence of comorbid anxiety disorders https://www.sciencedirect.com/science/article/abs/pii/S0278584611002430 and https://www.sciencedirect.com/science/article/abs/pii/S016517811200323X and https://onlinelibrary.wiley.com/doi/full/10.1002/brb3.1813 and https://www.sciencedirect.com/science/article/abs/pii/S0022510X06001043 ).
Please justify the reasons for the inclusion of these specific neuropsychological tests.
Were z scores used or raw scores in the analyses?
How was the age ranged selected? Please justify.
Please provide more details for the intervention.
Authors need to discuss the role of comorbid neurological diseases such as MS which also influences neuropsychological performance in BD patients exactly on these administered tests (https://onlinelibrary.wiley.com/doi/pdf/10.3233/BEN-2012-110198 and https://link.springer.com/article/10.1007/s00415-022-11359-6).
Effect sizes are missing in the statistics. Please add.
A more critical discussion of the findings regarding the clinical applications must be provided as a suggestion of how clinicians can include these findings in their practice.
Comments on the Quality of English LanguageA native English speaker should go through the text.
Author Response
Review 2
This is an interesting study examining the long-term impact of Cognitive Remediation (CR) interventions in BD patients. Some points for a major revision are below:
Thank you for your feedback and the time spent in helping improve the article. All modifications are highlighted in red in the text
How was the sample size estimated? Please add in the Methods section.
Thank you for your comment. The sample size was initially estimated based on a target of 60 participants, which was deemed appropriate to assess feasibility outcomes and provide preliminary data on the efficacy of the VR intervention. However, due to the early cessation of the enrollment phase caused by the COVID-19 pandemic, the final sample consisted of 50 participants. Given the limited evidence in this area of research, a precise sample size calculation for this specific study design was not feasible. Therefore, the primary aim of the study is to assess feasibility and gather preliminary efficacy data. A more detailed sample size calculation will be conducted in the efficacy study, based on the Phase II data from the current trial. We include this clarification in the Methods section.
Authors should discuss what that they did not include anxiety and depression questionnaires which heavily influence the neuropsychological performance of the participants with BD (for a relevant methodological comment on this to use about high anxiety levels: https://europepmc.org/article/med/28803143 as well as for existence of comorbid anxiety disorders https://www.sciencedirect.com/science/article/abs/pii/S0278584611002430 and https://www.sciencedirect.com/science/article/abs/pii/S016517811200323X and https://onlinelibrary.wiley.com/doi/full/10.1002/brb3.1813 and https://www.sciencedirect.com/science/article/abs/pii/S0022510X06001043 ).
Thank you for your comment, which allows us to further clarify these points. The suggested variables were measured in the previously published studies of the protocol and post-treatment results. It should be noted that, as this is a Phase 2 feasibility study, the primary outcome was the feasibility of the study, while the secondary outcomes included a preliminary evaluation of efficacy in terms of cognitive and psycho-social variables (personal and social functioning, levels of anxiety, depressive symptoms, and alexithymia, quality of life, biological and social rhythms). In the present study, the focus is instead on the secondary outcome of the preliminary measurement of efficacy in terms of cognitive functions at 6 and 12 months. This has been further clarified in the title and methods sections.
Please justify the reasons for the inclusion of these specific neuropsychological tests.
Thank you for your comment; this justification has been provided in the Methods section.
Were z scores used or raw scores in the analyses?
Z-Scores were adjusted using normative data corrected for age and education level, and equivalent scores (0-4) were used when available as normative data. To better clarify we added in the method section.
How was the age ranged selected? Please justify.
The established age range allows for secondary analyses on various age subgroups, and such analyses have been published elsewhere. To avoid potential confounding factors, we did not specify this in the text, as the present study did not focus on secondary analyses.
Please provide more details for the intervention.
We appreciate the comment, as it allows us to clarify certain aspects that are sometimes taken for granted but are important to make explicit for effective information dissemination.
Authors need to discuss the role of comorbid neurological diseases such as MS which also influences neuropsychological performance in BD patients exactly on these administered tests (https://onlinelibrary.wiley.com/doi/pdf/10.3233/BEN-2012-110198 and https://link.springer.com/article/10.1007/s00415-022-11359-6).
Thank you for your insightful comment. We appreciate the suggestion to discuss the role of comorbid neurological diseases, such as multiple sclerosis, and their influence on neuropsychological performance in BD patients. However, it is important to note that only 2 participants out of the final sample of 36 have a diagnosis of multiple sclerosis, which we believe does not significantly impact the findings of our study. We have previously published pre- and post-treatment analyses regarding subgroups with comorbidities in other studies. For the sake of clarity and focus, we chose not to include these additional considerations in the current article, as we wanted to keep the discussion centered on the integrated methodology used (psychoeducational methods within cognitive remediation) and the long-term results related to cognitive variables.
Effect sizes are missing in the statistics. Please add.
Effect sizes were not calculated because a non-parametric test was used due to the non-normal distribution of the sample
A more critical discussion of the findings regarding the clinical applications must be provided as a suggestion of how clinicians can include these findings in their practice.
Thank you for your valuable feedback. We understand the importance of discussing the clinical applications of our findings in a more critical manner. We have revised the conclusions to better reflect how clinicians might incorporate these findings into their practice.
Comments on the Quality of English Language. A native English speaker should go through the text.
Thank you for the suggestion, we have revised the language throughout the entire manuscript.
Round 2
Reviewer 2 Report
Comments and Suggestions for Authors
Most points raised in the previous round have been included. Please avoid the term subjects and use participants instead. About the two participants, authors need to discuss even briefly this point: Authors need to discuss the role of comorbid neurological diseases such as MS which also influences neuropsychological performance in BD patients exactly on these administered tests (https://onlinelibrary.wiley.com/doi/pdf/10.3233/BEN-2012-110198 and https://link.springer.com/article/10.1007/s00415-022-11359-6).
Comments on the Quality of English LanguageA native English speaker should go through the text.
Author Response
Review 2
Most points raised in the previous round have been included. Please avoid the term subjects and use participants instead. About the two participants, authors need to discuss even briefly this point: Authors need to discuss the role of comorbid neurological diseases such as MS which also influences neuropsychological performance in BD patients exactly on these administered tests (https://onlinelibrary.wiley.com/doi/pdf/10.3233/BEN-2012-110198 and https://link.springer.com/article/10.1007/s00415-022-11359-6).
Thank you for these suggestions. We have thoroughly revised the entire manuscript to improve the language and terminology. Additionally, we have incorporated the requested modifications into the discussion section, highlighted in green.
Round 3
Reviewer 2 Report
Comments and Suggestions for Authors
Most points raised by the reviewers have been included. One minor point to discuss is that important variables such as those included in a methodological comment (https://europepmc.org/article/med/28803143) are missing. This point raised by the mentioned article has to be added in the future designs of relevant interventions (especially anxiety levels should be tested as they influence cognitive performance in the presented neuropsychological tests). Overall, a very interesting article!
Comments on the Quality of English LanguageA native English speaker should check the final text.
Author Response
Most points raised by the reviewers have been included. One minor point to discuss is that important variables such as those included in a methodological comment (https://europepmc.org/article/med/28803143) are missing. This point raised by the mentioned article has to be added in the future designs of relevant interventions (especially anxiety levels should be tested as they influence cognitive performance in the presented neuropsychological tests). Overall, a very interesting article!
Thank you for these suggestions. We have thoroughly revised the entire manuscript to improve the language and terminology. Additionally, we have incorporated the requested modifications into the discussion section, highlighted in green.